# The Boombox:
# Visual Reconstruction from Acoustic Vibrations

**Boyuan Chen, Mia Chiquier, Hod Lipson, Carl Vondrick**
Columbia University
boombox.cs.columbia.edu

**Abstract:** Interacting with bins and containers is a fundamental task in robotics, making state estimation of the objects inside the bin critical. While robots often use cameras for state estimation, the visual modality is not always ideal due to occlusions and poor illumination. We introduce The Boombox, a container that uses sound to estimate the state of the contents inside a box. Based on the observation that the collision between objects and its containers will cause an acoustic vibration, we present a convolutional network for learning to reconstruct visual scenes. Although we use low-cost and low-power contact microphones to detect the vibrations, our results show that learning from multimodal data enables state estimation from affordable audio sensors. Due to the many ways that robots use containers, we believe the box will have a number of applications in robotics.

**Keywords:** Multimodal Perception, Object State Estimation, Audio

## 1 Introduction

In order for robots to robustly grasp and pick up objects inside containers, they need to accurately localize and estimate the state of the objects inside the container. Vision-based perception has enabled numerous advances for state estimation [1, 2, 3] in object manipulation and grasping, such as package bin-picking and object retrieval in household settings. Due to the importance of this problem, pose and state estimation from vision has been long studied in robot vision [4, 5, 6, 7, 8, 9].

However, environmental conditions are not always ideal for cameras. Although robots frequently need to interact with containers, objects inside containers often become occluded from cameras, making state estimation from vision impractical. In unconstrained settings, environments can also lack ideal lighting conditions, with limited visual signals to indicate the location and state of objects. Moreover, camera systems require extensive calibration for 3D state estimation, which is often fragile during contact and collisions.

In this paper, we propose to use *sound* as another sensing modality for object state estimation in robotics. Unlike vision, sound remains robust during occlusions or poor illuminations. For example, when an object is dropped inside a container, it may not be visible to any camera, but the collision between the object and the bin will cause sound that can be easily picked up by a contact microphone. The exact incidental vibration will depend on the location and pose of the objects inside the bin. We demonstrate how to use this audio signal to reconstruct the objects' states inside containers.

We introduce The Boombox, a smart container that uses the vibration of itself to reconstruct an image of its contents. The box is no larger than a cubic square foot. Unlike most containers, the box uses contact microphones to detect its own vibration. Exploiting the link between acoustic and visual structure, we show that a convolutional network can use these vibrations to predict the visual scene inside the container within centimeters, even under total occlusion and poor illumination. Figure 1 illustrates our box and one reconstruction from the sound.

The main contribution of this paper is an integrated hardware and software platform for using sound to reconstruct the visual structure inside containers. The remainder of this paper will describe The Boombox in detail. In section 2, we first review background on this topic. In section 3, we introduce our perception hardware, and in section 4, we describe our learning model. Finally, in section 5, we

5th Conference on Robot Learning (CoRL 2021), London, UK.

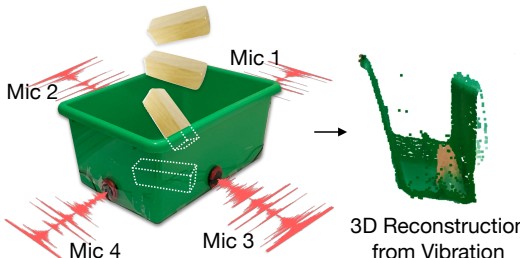

Figure 1: **The Boombox.** We introduce a "smart" container that is able to reconstruct an image of its inside contents. The approach works even when the camera cannot see into the container. The box has four contact microphones on each face. When objects interact with the box, they cause incidental acoustic vibrations. From these vibrations, we learn to predict the visual scene inside the box.

quantitatively and qualitatively analyze the performance and capabilities of our approach. We will open-source all hardware designs, software, models, and data.

## 2   Related Works

**Vision and Sound.** Our paper builds on work that integrates sound and visual perception. Given visual observations, there has been extensive studies to enhance sounds [10, 11], fill in missing sounds [12], and generate sounds entirely from video [13, 14]. Further, there have been recent works in integrating vision and sound to improve recognition of environmental properties [15, 16] and object properties, such as geometry and materials [17, 18]. Audiovisual data have also been studied for representation learning [19, 20]. Lastly, there has been work for generating a face given a voice [21] and a scene from ambient sound [22]. In contrast, our work uses sound to predict the 3D visual structure inside a container.

**Sound in Robotics.** Most related research to our work is on audiovisual object understanding for robot perception and control. Gandhi et al. [23] investigates vision and sound in a robot setting to predict which robot actions caused the given sound. Matl et al. [24] leverages audio signals from ball bouncing motions to calibrate the stochastic dynamical events for "sim2real" tasks. Bu and Huang [25] predicts the trajectory of a falling cube for robot object retrieval outside visible region. Instead of predicting the 2D trajectory for a single cube, this paper learns to predict the entire visual scene under fully occluded conditions by outputting RGB and depth images on three different blocks.

**Audio Analysis.** The primary features in audio that are used for sound localization [26] are time difference of arrival and level (amplitude) difference. Specifically calculating these exact features is non-trivial, especially in situations where the signal is not broad-band and in motion [27, 28, 29, 30]. Furthermore, these rough approximations can only be used to localize the object, whereas our goal is to not only localize objects, but also predict the 3D structure, which includes shape and orientation of the object. As such, we develop a model that learns the necessary features for reconstruction.

## 3   The Boombox

In this section, we present the The Boombox hardware. We also discuss the characteristics of the acoustic signals captured by it.

### 3.1   Detecting Vibrations

The Boombox, shown in Figure 2A, is a plastic storage container that is $15.5\text{cm} \times 26\text{cm} \times 13\text{cm}$ (width $\times$ length $\times$ height) with an open top. The box is a standard object that one can buy at any local hardware store. When an object collides with the box, a small acoustic vibration will be produced in both the air and the solid box itself. We have attached contact microphones on each wall of the plastic cuboid storage bin in order to detect this vibration. Unlike air microphones, contact microphones are insensitive to the vibrations in the air (which human ears hear as sound). Instead, they detect the vibration of solid objects.

The microphones are attached on the outer side of the walls, resulting in four audio channels. We arrange the microphones roughly at the horizontal center of each wall and close to the bottom. As

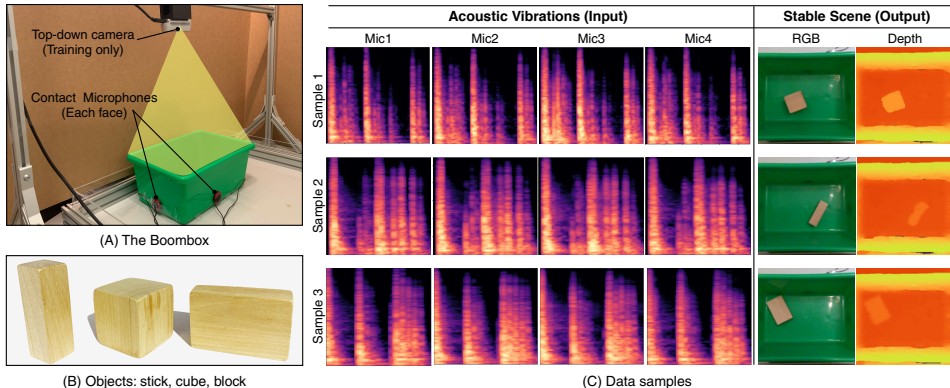

Figure 2: **The Boombox Overview.** (A) The Boombox can sense the object through four contact microphones on each side of a storage container. A top-down RGB-D camera is used to collect the final stabilized scene after the object movements. (B) We drop three wooden objects with different shapes. (C) Input and output data visualizations.

our approach will not require calibration, the microphone displacements can be approximate. We used TraderPlus piezo contact microphones, which are very affordable (no more than $5 each).[1]

## 3.2 Vibration Characteristics

When objects collide with the box, the contact microphones will capture the resulting vibrations. Figure 3 shows an example of the vibration captured from two of the microphones. We aim to recover the visual structure from this signal. As these vibrations are independent of the visual conditions, they allow perception despite occlusion and poor illumination. There is rich structure in the raw acoustic signal. For example, the human auditory system uses inter-aural timing difference (ITD), which is the time difference of arrival between both ears. Humans also locate sounds with inter-aural level difference (ILD), which is the amplitude level difference between both ears [31].

However, in our settings, extracting these characteristics is challenging. In practice, objects will bounce around in the container before arriving at their stable position, as shown in Figure 4. Each bounce will produce another, potentially interfering vibration. In our attempts to analytically use this signal, we found that the third bounce has the best signal for the time difference of arrival, but as can be seen from Figure 3, even on the third bounce the time difference of arrival is unclear in the actual waveform. There are a multitude of factors that make analytical approaches not robust to our real-world signals. Firstly, we are working with a moving signal, whereas time difference of arrival calculations work best on stationary signals due to the fact that it compares the time taken for a signal to travel from a fixed location. This makes it very difficult to analytically segment the signal into chunks of roughly the same location. Secondly, there are echos that make non-learning based methods difficult to identify phase shifts as the environment is a small container. Finally, the fact that the microphones are close together means that the time difference of arrival is encompassed in few samples, thus making it susceptible to noise.

Instead of hand crafting features, we will train a model to identify the fraction of the signal that is most robust for final localization. Our model will learn to identify the useful features from the signals to reconstruct a 3D scene, which includes the shape, orientation, and position of the contents.

## 3.3 Multimodal Training Dataset

Since vision and sound are naturally synchronized, we will use vision as self-supervision to learn robust characteristic features of the acoustic signal. We collected a multimodal training dataset by dropping objects into the box and capturing resulting images and vibrations. We position an Intel RealSense D435i camera that looks inside the bin to capture both RGB and depth images, which we

---

[1]We found that these microphones gave sufficiently clear signals while being more affordable than available directional microphone arrays. Each microphone was connected to a laptop through audio jack to USB converter. We use GarageBand software to record all four microphones together to synchronize the recordings.

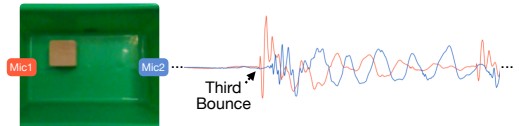

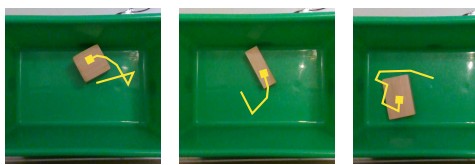

Figure 3: **Vibration Characteristics.** We visualize a vibration captured by two microphones in the box. Several distinctive characteristics need to be combined over time in order to accurately reconstruct an image with the right position, orientation, and shape of objects.

Figure 4: **Chaotic Trajectories.** We show the chaotic trajectories of objects as they bounce around the box until becoming stable. The moving sound source and multiple bounces also create potentially interfering vibrations, complicating the real-world audio signal.

only use during training.[2] We use three wooden blocks with different shapes to create our dataset. The blocks have the same color and materials, and we show these objects in Figure 2B. We hold the object above the bin, and freely drop it. After dropping, the objects bounce around in the box a few times before settling into a resting position. We record the full process from all the microphones and the top-down camera. Overall, our collection process results in diverse falling trajectories across all shapes with a total of 1,575 sequences. Figure 2C shows an overview of the dataset. After learning, our approach will be able to reconstruct the 3D visual scene from the box's vibration alone.

## 4 Predicting the Visual Scene from Vibration

To estimate the state inside the container, we will learn to reconstruct the visual modality from the audio modality. We present a convolutional network that translates vibrations into images.

### 4.1 Model

We will fit a model that reconstructs the visual contents from the vibrations. Let $A_i$ be a spectrogram of the vibration captured by microphone $i$ such that $i \in \{1, 2, 3, 4\}$. Our model will predict the image $\hat{X}_{\text{RGB}} = f_{\text{RGB}}(A; \theta)$ where $f$ is a neural network parameterized by $\theta$. The network will learn to predict the image of a top-down view into the container. We additionally have a corresponding network to produce a depth image $\hat{X}_{\text{depth}} = f_{\text{depth}}(A; \theta)$.

Reconstructing a pixel requires the model to have access to the full spectrogram. However, we also want to take advantage of the spatio-temporal structure of the signal. We therefore use a fully convolutional encoder and decoder architecture. The network transforms a spectrogram representation (time $\times$ frequency) into an $C$-channel embedding with width and height being $1 \times 1$, such that the receptive field of every dimension reaches every magnitude in the input and every pixel in the output. Unlike image-to-image translation problems [32, 33, 34], our task requires translation across modalities.

We use a multi-scale decoder network [35, 36, 37]. Specifically, each decoder layer consists of two branches. One branch is a transposed convolutional layer to up-sample the intermediate feature. The other branch passes the input feature first to a convolutional layer and then a transposed convolution so that the output for the second branch matches the size of the first branch. We then concatenate the output from these two branches along the feature dimension as the input feature for the next decoder layer. We perform the same operation for each decoder layer except the last layer where only one transposed convolution layer is needed to predict the final output image.

We use a spectrogram to represent audio signals. We apply a Fourier Transform before converting the generated spectrogram to Mel scale. Since we have four microphones, audio clips are concatenated together along a third dimension in addition to the original time and frequency dimension.

### 4.2 Learning

In most cases, we care about predicting the resting position of the object. We therefore train the network $f$ to predict the final stable image. For RGB image predictions, we train the network to

---

[2]The camera is 42cm away from the bottom of the bin to capture clear top-down images.

minimize the expected squared error between the image $X_{\text{RGB}}$ and the predictions from audio $A$:

$$\mathcal{L}_{\text{RGB}} = \mathbb{E}_{A,X} \left[ \| f_{\text{RGB}}(A;\theta) - X_{\text{RGB}} \|_2^2 \right] \tag{1}$$

In order to reconstruct shape, we also train the network to predict a depth image from the audio input. We train the model to minimize the expected L1 distance:

$$\mathcal{L}_{\text{depth}} = \mathbb{E}_{A,X} \left[ \| f_{\text{depth}}(A;\phi) - X_{\text{depth}} \|_1 \right] \tag{2}$$

Since ground truth depth often has outliers and substantial noise, we use an L1 loss [38]. We use stochastic gradient descent to estimate the network parameters $\theta$ and $\phi$. After learning, the model predicts both the RGB image and the depth image from just the vibration. The visual modality is only supervising representations for the audio modality, allowing reconstructions when cameras are not viable, such during occlusions or low illumination.

### 4.3 Implementation Details

Our network takes in the input size of $128 \times 128 \times 4$ where the last dimension denotes the number of microphones. The output is a $128 \times 128 \times 3$ RGB image or a $128 \times 128 \times 1$ depth image. We use the same network architecture for both the RGB and depth output representations except the feature dimension in the last layer for different modalities. All network details are listed in the Appendix. Our networks are configured in PyTorch [39] and PyTorch-Lightning [40]. We optimized all the networks for 500 epochs with Adam [41] optimizer and batch size of 32 on a single NVIDIA RTX 2080 Ti GPU. The learning rates starts from 0.001 and decrease by 50% at epoch 20, 50, and 100.

## 5 Experiments

Our experiments analyze the hardware and software at reconstructing an image of the contents from audio. In this section, we first quantitatively evaluate the performance. We then show qualitative results for the reconstructions. Finally, we visualize the learned representations.

Since the physics behind our dataset is chaotic, everytime an object is dropped into the container, we obtain a unique example with a different resting position and orientation. We randomly split the dataset into a training set (80%), a validation set (10%), and a testing set (10%). All of our results are evaluated with three random seeds for training and evaluation with various splits of the dataset. We report the mean and the standard error of the mean for all outcomes.

### 5.1 Evaluation Metrics

We use two evaluation metrics for our final scene reconstruction that focus on the object state.

**IoU** measures how well the model reconstructs both shape and location. Since the model predicts an image, we subtract the background to convert the predicted image into a segmentation mask. Similarly, we performed the same operation on the ground-truth image. IoU metric then computes intersection over union with the two binary masks.

**Localization** score evaluates whether the model produces an image with the block in the right spatial position. This metric is especially useful for object picking tasks with a suction gripper where the spatial location of the block matters. With the binary masks obtained in the above process, we can fit a bounding box with minimum area around the object region. We denote the distance between the center of the predicted bounding box and the center of the ground-truth bounding box as $d$, and the length of the diagonal line of ground-truth box as $l$. We report the fraction of times the predicted location is less than half the diagonal: $\frac{1}{N} \sum_{i=1}^{N} [d_i \leq l/2]$.

### 5.2 Baselines

**Time Difference of Arrival (TDoA).** We compare against an analytical prediction of the location. A standard practice is to localize sound sources by estimating the time difference of arrival across an array of microphones. In our case, the microphones *surround* the sound source. There are several ways to estimate the time difference of arrival, and we use the Generalized Cross Correlation with Phase Transform (GCC-PHAT), which is a established, textbook approach [42]. Once we have our

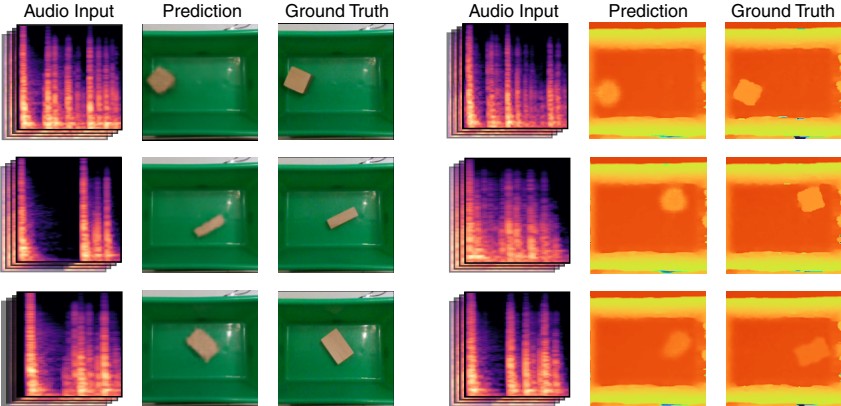

Figure 5: **Model Predictions with Mixed Shapes.** From left to right on each column, we visualize the audio input, the predicted scene, and the ground-truth images. Our model can produce accurate predictions for object shape, position and orientation.

time difference of arrival estimate, we find the location in the box that would yield a time difference of arrival that is closest to our estimate.

**Random Sampling.** To evaluate if the learned models simply memorize the training data, we compared our method against a random sampling procedure. This baseline makes a prediction by randomly sampling an image from the training set and using it as the prediction.

**Average Bounding Box.** The average bounding box baseline measures to what extent the model learns the dataset bias. We extracted object bounding boxes from all the training data through background subtraction and rectangle fitting to obtain the average center location, box sizes and box orientation. This baseline uses the average bounding box as the prediction for all the test samples.

**Nearest Neighbor.** To evaluate the generalization performance from training data distribution, we construct a nearest neighbor baseline. For each test input audio, we use the resulted image from the training data with most similar audio as the prediction. The similarity is measured by a L2 distance.

## 5.3 Reconstruction with Mixed Shapes

We first analyze how well The Boombox reconstructs its contents when the shape is not known a priori. We train a single model with all the object shapes. The training data for each shape are simply combined together so that the training, validation and testing data are naturally well-balanced with respect to the shapes. This setting is challenging because the model needs to learn audio features for multiple shapes at once.

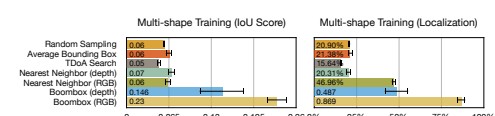

Figure 6: **Reconstruction with Mixed Shapes.** Our model outperforms baseline methods at localization and shape prediction.

Figure 6 shows the convolutional networks are able to learn robust features to localize both the position and orientation, even when shapes are mixed. Our method outperforms TDoA often by significant margins, suggesting that our learning-based model is learning robust acoustic features for localization. Due to the realistic complexity of the audio signal, the hand-crafted features are hard to reliably estimate. Our model outperforms both the random sampling and average bounding box baseline, indicating that our model learns the natural correspondence between acoustic signals and visual scene rather than memorizing the training data distribution. We show qualitative predictions for both RGB and depth images in Figure 5.

## 5.4 Reconstruction with Known Shape

We next analyze how well the model performs when the object shape is known, but the position and orientation is not. We train separate models for each shape of the object independently. Figure 7 shows The Boombox is able to reconstruct both the position and orientation of the shapes. The convolutional network obtains the best performance for most shapes on both evaluation metrics.

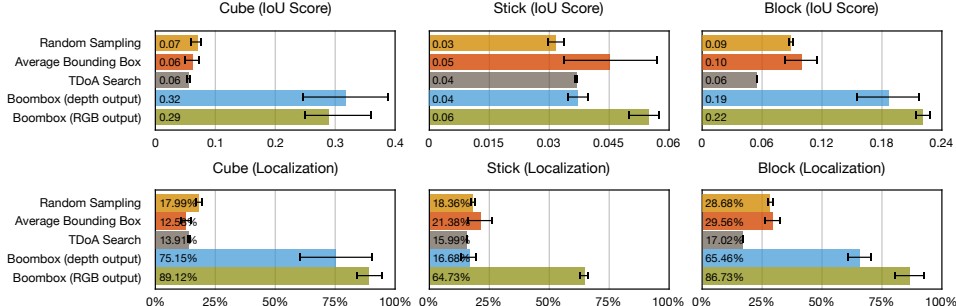

Figure 7: **Reconstruction with Known Shape.** We show the performance of each individual model trained with one of the three objects. We report both the mean and the standard error of the mean from three random seeds. Our approach enables robust features to be learned to predict the location and shape of the dropped objects.

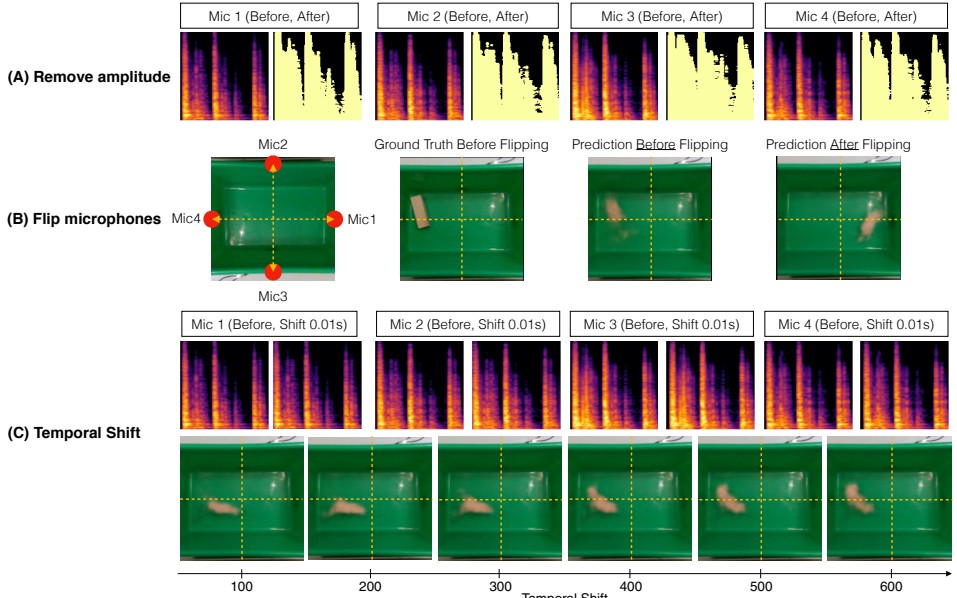

Figure 8: **Visualization of Ablation Studies.** We visualize the impact of different ablations on the model. **A**) By thresholding the spectrograms, we remove the amplitude from the input. **B**) We experimented with flipping the microphones only at testing time. The model's predictions show a corresponding flip as well in the predicted images. **C**) We also experimented with shifting the relative time difference between the microphones, introducing an artificial delay in the microphones only at testing time. A shift in time causes a shift in space in the model's predictions. The corruptions are consistent with a block falling in that location.

These results highlight the relative difficulty at reconstructing different shapes from sound. By comparing the model performance across various shapes, the model trained on cubes achieves the best performance while the model trained on blocks performs slightly worse. The most difficult shape is the stick.

When the training data combines all shapes, the model should share features between shapes, thus improving performance. To validate this, we compare performance on the multi-shape versus models trained with a single known shape. Figure 9 shows that the performance on the block and stick shapes are improved by a large margin. We notice that the performance of the cube drops due to the confusion between

| | RGB output (IoU Score, single) | RGB output (IoU Score, single, error) | Depth output (IoU Score, single) | Depth output (IoU Score, single, error) |
|---|---|---|---|---|
| | 0.05514 | 0.0052831247382586 | 0.0373403333333333 | 0.002613145 |
| | 0.221466 | 0.0075120207889312 | 0.186803 | 0.031363496 |

Figure 9: **Shape Transfer.** Performance improves by training with multiple shapes.

shapes. When the cube confuses with the stick or the block, because of the smaller surface area of these two shapes, the cube performance slightly degrades.

## 5.5 Ablations and Analysis

To better understand what features the model has learned specifically, we perform several ablation studies, shown in Figure 8 and Figure 10.

**Flip microphones.** The microphones' layout should matter for our learned model to localize the objects. When we flipped the microphone location, due to the symmetric nature of the hardware setup, the predictions should also be flipped accordingly. To study this, we flipped the input of Mic1 and Mic4 as well as the input of Mic2 and Mic3 in the testing set, shown in Figure 8. Our results in Figure 8B shows that our model indeed produces a flipped scene. The performance in Figure 10 nearly drops to zero, suggesting that the model implicitly learned the relative microphone locations to assist its final prediction.

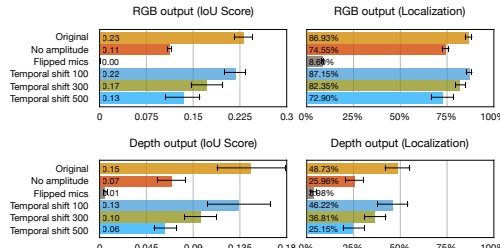

Figure 10: **Quantitative ablation studies.** We experiment with different perturbations to our input data to understand the model predictions.

**Remove amplitude.** The relative amplitude between microphones can indicate the relative position of the sound source to different microphones. We removed the amplitude information by thresholding the spectrograms, shown in Figure 8. We retrained the network due to potential distribution shift. As expected, even though the time and frequency information are preserved, the model performs much worse (Figure 10), suggesting that our model additionally learns to use amplitude for the predictions.

**Temporal shift.** We are interested to see if our model learns to capture features about the time difference of arrival between microphones. If so, when we shift the audio temporally, the prediction should also shift spatially. We experimented with various degrees of temporal shifts on the original spectrograms. For example, shifting 500 samples corresponds to shifting about 0.01s (500 / 44,000). By shifting the Mic1's spectrogram forward and Mic4's spectrogram backward with zero padding to maintain the same amount of time, and preforming similar operation on Mic2 and Mic3 respectively, we should expect that the predicted object position shifts towards the left-up direction. In Figure 8, we can clearly observe this trend as temporal shift increases. Shifting the signal in time decreases the model's performance, demonstrating that the model has picked up on the time difference of arrival.

**Feature Visualization:** We finally visualize the latent features in between our encoder and decoder network with t-SNE[43], shown in Figure 11. We colorize the points based on ground truth position and orientation. The magnitude distance from the center of the image is represented by saturation, and the angle from the horizontal axis is represented by hue. We find that there is often clear clustering of the embeddings by their position and orientation, showing that the model is robustly discriminating the location of the impact from sound alone. Moreover, the gradual transitions between colors suggest the features are able to smoothly interpolate spatially.

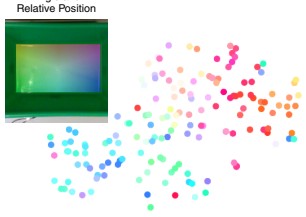

Figure 11: **Low-dimensional embedding.** We visualize the learned features in the encoder with t-SNE.

## 6 Conclusion

This paper introduces The Boombox, a low-cost container integrated with a convolutional network that uses sound to reconstruct an image of the contents inside it. Containers are ubiquitous in everyday situations, and this paper shows that we can equip them with sound perception to estimate the state inside the container. When cameras are not available, such as during occlusions or poor illumination, our results suggest that sound is a promising sensing modality.

In the future, one exciting direction is to leverage the acoustic signatures to not only infer the 3D geometries of the object, but also the materials of various objects. In multi-object interaction setting such as collision, audio can also be an important sensing signal to understand their dynamics. We also aim to scale our current object collections to include diverse objects.

**Acknowledgments**

We thank Philippe Wyder, Dídac Surís, Basile Van Hoorick and Robert Kwiatkowski for helpful feedback. This research is based on work partially supported by NSF NRI Award #1925157, NSF CAREER Award #2046910, DARPA MTO grant L2M Program HR0011-18-2-0020, and an Amazon Research Award. MC is supported by a CAIT Amazon PhD fellowship. We thank NVidia for GPU donations. The views and conclusions contained herein are those of the authors and should not be interpreted as necessarily representing the official policies, either expressed or implied, of the sponsors.

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
