# OpenReview forum: "The Boombox: Visual Reconstruction from Acoustic Vibrations"
_robot-learning.org/CoRL/2021/Conference — CoRL2021 Poster_

### Official Review · Reviewer_aA4p · 2021-07-09

**Originality:** Very Good
**Technical Quality:** Very Good
**Clarity Of Presentation:** Good
**Impact:** 4

**Recommendation:**

Strong Accept: I recommend accepting the paper and will argue for my recommendation even if other reviewers hold a different opinion.

**Summary:**

This paper proposes a boombox, a container that can estimate the location of objects thrown into the box from an array of microphones. The audio signal is processed by a convolutional network to reconstruct the RGB and depth image of the objects from a bird's eye view. The approach is validated over multiple setups and shown to outperform traditional audio processing methods.

**Issues:**

I think that during the rebuttal period some simple experiments could be done to make the contribution stronger. Specifically:
1. Predict directly the object location or segmentation instead of the depth or rgb image.
2. Include the nearest neighbor baseline explained above.
In addition, it would be nice to see the comments on the presentation to be included.

**Reviewer Expertise:**

Good: General knowledge of the area

**Strengths And Weaknesses:**

Overall, I find this paper to be very interesting. The idea of reconstructing the visual signal from only audio is very intriguing and novel (to the best of my knowledge), and I am surprised that it works so well. Much related work on audio-visual processing exists (as pointed out in the related work), but I think this work stands out from existing methods from both the problem setup and the results. Along these lines, it would have been great to read a discussion in related work about other methods of non-line-of-sight shape reconstruction, e.g. [1] or [2].

The approach is clearly presented and well-executed, with in-depth experimental analysis to understand how the approach compares to traditional methods and what features in the audio signal are used for reconstruction.

However, there are still some issues and points to clarify to make the contribution even stronger.
From a methodological point of view, it is unclear why learning an RGB image from audio. Clearly, no color information is available in the audio, and the results are in "color" just because of the overfit to the datasets' color. What can be done to solve this issue (and possibly favor generalization to other boxes/objects)? I would assume that it would be much easier to learn an "intensity-free" image (e.g., depth) or some low-level feature (e.g., the x-y coordinate of the object or the object segmentation) instead of an RGB or grayscale image. Along these lines, I am very surprised that reconstructing a depth works consistently worse than the RGB. Why is this the case? What are the challenges that are present for depth but absent in RGB? Therefore, I would be very interested in seeing a more detailed discussion about the topic and possibly seeing some quantitative results of directly regressing the object segmentation and bounding box from the audio signal. In addition, it would be nice to see how other state-of-the-art architectures for audio processing, e.g., wavenet and its more recent successors, perform compared to the simple proposed encoder-decoder architecture.

Finally, I think that the presentation could be improved in some places. More specifically:
1. The discussion in the intro on why visual sensors may fail reads a bit weak. I believe that "occlusions" in the box could also be a problem for the proposed method (since a completely different audio signal would be generated). Also, motion blur does not seem relevant to the task since one would always wait for the objects to land in a stable position. Maybe some stronger motivation can be found (e.g., redundancy with a vision sensor?)
2. Some figures (e.g., Fig 6 and 10) are tiny and difficult to read.
3. Paragraph 5.3 (reconstruction with unknown shape) is a bit deceiving. The experiment does not show what happens with shapes that are outside the training dataset. I would therefore recommend changing the title of the section to "Reconstruction with mixed shapes" or something along these lines. That being said, it would actually be nice to see some experiments where reconstruction of previously unseen shapes is studied (I guess it would not work, but to which extent?)
4. The "Random" baseline is very weak and uninformative. I would substitute it with a stronger one that returns the label associated to the nearest neighbor (according to some simple metric) of the training dataset.
5. There is no discussion about the limitations of the approach. I think this is fundamental to enable researchers to follow up the work.



[1] A Theory of Fermat Paths for Non-Line-of-Sight Shape Reconstruction, Xin et al.
[2] Acoustic Non-Line-of-Sight Imaging, Lindell, et al.

**Summary Of Recommendation:**

I mainly base my recommendation on the novelty of the idea and the interesting results.

---

> ### Author Response · Authors · 2021-08-28
> **Responses to Reviewer aA4p (Part 2)**
>
> **“Paragraph 5.3 (reconstruction with unknown shape) is a bit deceiving. The experiment does not show what happens with shapes that are outside the training dataset. I would therefore recommend changing the title of the section to "Reconstruction with mixed shapes" or something along these lines. That being said, it would actually be nice to see some experiments where reconstruction of previously unseen shapes is studied (I guess it would not work, but to which extent?)”**
>
> Thank you for pointing this out. We will update the paper with “mixed shapes”. We would love to experiment with various scenarios with different objects. However, we would like to emphasize that the current dataset is already quite challenging to study as is. As shown in our results, traditional sound engineering techniques suffer a lot under the challenging setup (fast motion, chaotic trajectory, small bin with echo, etc. As laid out in Section 3.2.). Even with our learning-based approaches, the scores vary among different shapes with the overall localization percentage being 87%. It requires more research efforts to even solve the current dataset. Therefore, we believe large-scale studies on more objects deserve several iterations of future works along the direction. We hope this paper will spur more research on sound inside the robotics community.
>
> Moreover, collecting the dataset with the current setup is not trivial and already reveals useful conclusions for future studies. Specifically, there are several challenges and considerations during the dataset collection:
>
> - The dataset was collected under the restrictions of the COVID-19 pandemic where we do not have a lab environment. To obtain a controlled dataset (more on this below) for rigorous research, we had to overcome multiple challenges to construct the dataset.
>
> - We have collected more than 500 data points for each object shape. In total, there are more than 1,500 data points. This is already non-trivial for real-world data collection which costs about 15 days of data collection, the setup inspection, and manual data quality inspection. We have manually inspected all the 4-channel audios, RGB images, as well as depth images.
>
> - Challenges of collecting real-world well-controlled datasets include:
>   - The quality of the audio could be affected by the ambient sound. To obtain controlled audio signals, the environment needs to be quiet enough. We also need to consider the impact of the noise made by the dropping motion on other people in the neighborhood, so we could only collect the data within certain hours during quiet periods.
>   - The humidity of the wooden objects might affect the sound. We avoided this by wearing disposable gloves every time we started the data collection.
>   - To avoid the bias of lightning conditions on the ground truth RGB-D images, we needed to control the lighting conditions.
>   - The table surface holding the bin needs to be maintained at the same level to avoid potential bias introduced by the table level. We carefully calibrate the table surface every time before data collection. The cameras also need to be fixed to avoid bias introduced into the dataset.
>   - The color of the wooden objects may fade over a long time period, so the data collection has to happen within a short timeframe.
>   - Other challenges could affect the stationarity of the data collection: the bin might deform over time due to dropping motions, mics contact properties might change over time, the bin might move during impact. All of these need to be verified during the collection process.
>   - There is a crucial factor: the fact that humans need to drop the object to simulate real-world scenarios limits the possibility to automate the data collection.
>
> **“The "Random" baseline is very weak and uninformative. I would substitute it with a stronger one that returns the label associated with the nearest neighbor (according to some simple metric) of the training dataset.”**
>
> Thank you for your great suggestion. We are happy to report that we have rerun the evaluation with this nearest neighbor baseline on both RGB and depth output. Specifically, Figure 6 has been updated to incorporate the new results. The nearest neighbor approach performs similar to other baselines or slightly better than other baselines for certain random seeds, while our method still performs much better. Therefore, our conclusion still holds.
>
> **“There is no discussion about the limitations of the approach. I think this is fundamental to enable researchers to follow up the work.”**
>
> Thank you for pointing this out. We will update the paper with discussions on limitations and future works.

---

> ### Author Response · Authors · 2021-08-28
> **Responses to Reviewer aA4p (Part 1)**
>
> Thank you for your suggestions and the appreciation of our work. We hope to address your questions with the following point responses.
>
> **“Along these lines, it would have been great to read a discussion in related work about other methods of non-line-of-sight shape reconstruction.”**
>
> Thank you for your suggestions. We will update the paper to include these discussions.
>
> **“From a methodological point of view, it is unclear why learning an RGB image from audio. Clearly, no color information is available in the audio, and the results are in "color" just because of the overfit to the datasets' color. What can be done to solve this issue (and possibly favor generalization to other boxes/objects)? I would assume that it would be much easier to learn an "intensity-free" image (e.g., depth) or some low-level feature (e.g., the x-y coordinate of the object or the object segmentation) instead of an RGB or grayscale image. Along these lines, I am very surprised that reconstructing a depth works consistently worse than the RGB. Why is this the case? What are the challenges that are present for depth but absent in RGB? Therefore, I would be very interested in seeing a more detailed discussion about the topic and possibly seeing some quantitative results of directly regressing the object segmentation and bounding box from the audio signal. In addition, it would be nice to see how other state-of-the-art architectures for audio processing, e.g., wavenet and its more recent successors, perform compared to the simple proposed encoder-decoder architecture.”**
>
> In the real-world setup, perceiving depth information with commonly available camera sensors is much more challenging than RGB images. The real-world depth image is very noisy as seen in the examples where the boundaries of the objects are often not super clear. This introduces more confusion when different shapes are mixed together. Moreover, the bin is made of plastics. The strong reflection material property of it also introduces more noises into the depth image. We believe that the noisy ground-truth data introduces extra challenges for the model to learn a consistent representation.
>
> There are several advantages to predict RGB and depth images. First, they come with strong interpretation capabilities. Second, they are a more complete representation. In different downstream tasks, they can be used to extract different information that matters, such as segmentation, center position, shape, object orientation, etc. Furthermore, one recent promising planning research direction is to use images as goals for planning and control. Predicting in more complete representation enables the flexible usage of the predictions in various formats. Last, we do not assume access to human labeling for the objects due to generality purposes. Practically, it is often challenging and costly to obtain accurate segmentation labels. The x-y coordinates often miss critical factors for downstream tasks such as the shape and rotation of the object.
>
> State-of-the-art audio processing architectures such as WaveNet require huge amounts of data. In our scenario, obtaining large-scale real-world data is very difficult and expensive. The advantage of our approach is that our model can work well with small amounts of data. Another advantage of our model is that it is lightweight. Therefore, for applications in robotics where real-time prediction and feedback are important, our model can provide real-time responses (188 Hz with one GPU).
>
> **“The discussion in the intro on why visual sensors may fail reads a bit weak. I believe that "occlusions" in the box could also be a problem for the proposed method (since a completely different audio signal would be generated). Also, motion blur does not seem relevant to the task since one would always wait for the objects to land in a stable position. Maybe some stronger motivation can be found (e.g., redundancy with a vision sensor?)”**
>
> Thank you for your suggestions. What we mean by “occlusions” is the sensing modality is being occluded. The camera will not be able to see anything while the wall of the bin is in between the object and the camera. When there are occlusions in the box, exactly as the reviewer suggested, the audio signal can be different which is exactly the underlying mechanism of our appraoch. The generated audio signals can still be captured by our devices. Therefore, with appropriate training data, the acoustic signatures can still be recognized instead of completely missing. This is an interesting future direction, but currently out of the scope of our study.
>
> Thanks for the suggestions on the motion blur, we removed the argument in the paper.
>
> **“Some figures (e.g., Fig 6 and 10) are tiny and difficult to read.”**
>
> We will update the paper with larger pictures.

---

> > ### Comment · Reviewer_aA4p · 2021-09-01
> > **Regarding the output modality**
> >
> > Thanks for your answers and for adding additional experiments. What I am still a bit confused about is the argument on why predicting RGB is easier than depth (or even grayscale). I still don't see how the intensity (and in particular the color) information would be observable from audio signal. I do not agree with the current reasoning that the problem lies in the noise of depth observations. This factor can be normalized away by computing depth from images using either triangulation or a SOTA neural network (for example [1]), instead of using the RealSense output. This experiment would be easy to run with the existing dataset, as far as I understand. Could maybe the authors comment on this?
> >
> >
> > [1] Towards Robust Monocular Depth Estimation: Mixing Datasets for Zero-shot Cross-dataset Transfer, Ranftl et al.

---

> > > ### Author Response · Authors · 2021-09-01
> > > **Response to Reviewer aA4p**
> > >
> > > Thanks for your follow-up questions. We hope to answer your follow-up questions with the following point responses.
> > >
> > > **I still don't see how the intensity (and in particular the color) information would be observable from audio signal.**
> > >
> > > The network is able to learn the color and intensity information because the neural network is trained in a closed environment where the color of the bin and objects remain relatively same across different initial conditions. Therefore, the neural network can pick these environmental biases from the training data while generalizing on various unseen dropping conditions.
> > >
> > > **What I am still a bit confused about is the argument on why predicting RGB is easier than depth (or even grayscale). I do not agree with the current reasoning that the problem lies in the noise of depth observations. This factor can be normalized away by computing depth from images using either triangulation or a SOTA neural network (for example [1]), instead of using the RealSense output.**
> > >
> > > - We indeed experimented with various ways on processing the depth image, such as smooth filters, camera distances, lightning conditions, and filling holes. We made sure to produce the depth image as best as we can. As shown in the paper, the depth image matches the SOTA quality with the available sensors. However, the depth images are still nosier than RGB images caused by reflections of the plastic bin and the hardware noises. This is still not a solved challenge that requires both hardware and software research. With better depth sensing capabilities, we believe that performance can be further improved.
> > >
> > > - We would like to emphasize that predicting depth information is fundamentally more challenging than predicting RGB information. Predicting depth information requires the understanding of 3D spatial knowledge while predicting RGB information only asks for 2D signals. Scaling from 2D to 3D learning is not trivial, hence the performance gap.g
> > >
> > > - Computing depth from images using triangulation requires multiple cameras and careful camera calibration, while our setup do not require any of these. Modifying the hardware to include more cameras and careful calibration add non-trivial human efforts and higher costs to the system. The fundamental challenge of predicting 3D information vs. 2D still persists.
> > >
> > > - We thank the reviewer for the pointer of SOTA neural network methods to estimate depth image from monocular image. We experimented with the best model from the framework and found that the results are consistently much worse than our ground truth depth images. For example, the framework gives wrong depth estimation for the bottom of the bin with a value around 13.5 cm while the ground-truth measurement is around 42 cm. Our depth images matches this measurement. Another example is that the depth estimation does not remain consistent even on the flat surface of the object.  We suspect that the inferior performance is due to the following two reasons. First, the training and testing dataset used in the pointed paper belong to very different domains than our scenario. Second, our dataset comes with low texture information while the examples shown in the pointed paper work well on natural open world images with highly textured objects and scenes. Consequently, the pointed framework is not directly applicable to our method.

---

> > > > ### Comment · Reviewer_aA4p · 2021-09-03
> > > > **Thanks for the answer**
> > > >
> > > > I totally see that in a single dataset predicting RGB could be helpful and easier in than depth (with noise). However, I think this could be a real problem towards generalization. Just imagine what would happen if the color of the background or the box would change... For this reason, I was trying to look for something that could simplify generalization. I still think depth (maybe with more changes or tuning) could do the job.
> > > > Overall, this could be a nice research direction for the future, and I am happy to support acceptance. However, I would recommend adding generalization as one of the main limitations of the current method (as also other reviewers pointed out). This would facilitate future work.
> > > >
> > > > PS: The monocular depth estimation method I recommended can only produce depth **up to scale**. Therefore, you would need to normalize it for the per-image minimum to get consistent results. Also, using something as inverse depth instead of depth could probably help.

---

### Official Review · Reviewer_5sJt · 2021-07-19

**Originality:** Good
**Technical Quality:** Good
**Clarity Of Presentation:** Very Good
**Impact:** 3

**Recommendation:**

Weak Accept: I recommend accepting the paper, but will not argue for my recommendation if the majority of other reviewers have a different opinion.

**Summary:**

This paper introduces a hardware setup called Boombox and an algorithm that can predict visual signals from acoustic vibrations.
Specifically, the box is equipped with four contact microphones, one on each side, to provide audio signals. The learning algorithm is trained to predict the RGB or depth image of the final scene of the inside of the box after a wooden object gets dropped into it.
This system is shown to work more reliably than the alternative methods considered, qualitatively and quantitatively under the two metrics IOU and localization. The three alternative methods are time difference of arrival (TDoA) using Generalized Cross Correlation with Phase Transform(GCC-PHAT), random sampling from the training set, average bounding box from the training set.

**Issues:**

Please refer to the three items discussed in the weaknesses section.

**Reviewer Expertise:**

Fair: Some knowledge of the area

**Strengths And Weaknesses:**

Main strengths:
1. The paper is well written
2. The authors built a system that predicts the 3D visual scene from audio signals and showed that it worked well under the setting described
3. Ablation studies were thorough: flipped microphones, removed amplitude, temporal shift. The learned features were also shown to have learned spatial information.

Weaknesses / Suggestions:
1. It’s not clear to me that this type of approach would scale under the situation of object collisions, more realistic object types:
a. The objects that were included in the experiments only differed in shape. They shared the same color and materials.
b. The paper only considered dropping one object at a time into an empty box. It did not discuss whether this approach can be adapted to handle more complex settings when there're collisions between objects.
While the above two points might be too much to ask in a pilot study, I was hoping that they would be discussed in the paper. If the authors consider that to be future works, I would suggest discussing it in the paper as part of the limitation of the method.

2. I feel that there may be another baseline that this paper did not experiment with. In the baseline method “random sampling”, one image from the training set is chosen as the prediction of the test audio sample. One could imagine a nearest neighbor baseline that finds the nearest neighbor match from the training set of the test audio. This new baseline would indicate the similarity between the train and test distributions.

3. Background subtraction seems to be an important step in computing both evaluation metrics since the error in the background subtraction output would affect the quality of downstream tasks (during test time). Yet, neither the method of background subtraction nor its impact was described.


**Summary Of Recommendation:**

The task of using sound to help to estimate object state is interesting.
The paper describes a system that works in relatively simple settings where a single object is dropped into an empty bin.
It's well written and the method shows some real-world success.
Hence my overall impression of the paper is positive. But I hold some concerns about the practicality of the approach. In real-world tasks where there are a larger variety of objects and with the addition of object-object collision, using acoustic vibration alone to estimate the object state seems difficult.

---

> ### Author Response · Authors · 2021-08-28
> **Responses to Reviewer 5sJt (Part 1)**
>
> Thank you for your suggestions and the appreciation of our work. We hope to use the following point response to resolve your questions.
>
> **“It’s not clear to me that this type of approach would scale under the situation of object collisions, more realistic object types: a. The objects that were included in the experiments only differed in shape. They shared the same color and materials. b. The paper only considered dropping one object at a time into an empty box. It did not discuss whether this approach can be adapted to handle more complex settings when there're collisions between objects. While the above two points might be too much to ask in a pilot study, I was hoping that they would be discussed in the paper. If the authors consider that to be future works, I would suggest discussing it in the paper as part of the limitation of the method.”**
>
> We would like to emphasize that the current dataset is already quite challenging to study as is. As shown in our results, traditional sound engineering techniques suffer a lot under the challenging setup (fast motion, chaotic trajectory, small bin with echo, etc. As laid out in Section 3.2.). Even with our learning-based approaches, the scores vary among different shapes with the overall localization percentage being 87%. It requires more research efforts to even solve the current dataset. Therefore, we believe large-scale studies on more objects deserve several iterations of future works along the direction. We hope this paper will spur more research on sound inside the robotics community.
>
> We agree with the reviewer that these are great directions to study for future works. We will update the paper to discuss this as future studies and limitations.
>
> **“I feel that there may be another baseline that this paper did not experiment with. In the baseline method “random sampling”, one image from the training set is chosen as the prediction of the test audio sample. One could imagine a nearest neighbor baseline that finds the nearest neighbor match from the training set of the test audio. This new baseline would indicate the similarity between the train and test distributions.”**
>
> Thank you for pointing out this baseline. We are happy to report that we have rerun the evaluation with this baseline on both RGB and depth output. Specifically, Figure 6 has been updated to incorporate the new results. The nearest neighbor approach performs similar to other baselines or slightly better than other baselines for certain random seeds, while our method still performs much better. Therefore, our conclusion still holds.
>
> **“Background subtraction seems to be an important step in computing both evaluation metrics since the error in the background subtraction output would affect the quality of downstream tasks (during test time). Yet, neither the method of background subtraction nor its impact was described.”**
>
> Thank you for pointing this out. The background subtraction takes the color range of the target object and finds the largest area with the same color range to obtain the mask for subtraction. We tuned the color range for the target object on a single training data point per shape and used it as is for all the evaluations. We have manually inspected the mask generated with this one-time tuned parameter. We found that it works robust and consistently for all the rest of the examples without any outlier. We will include the algorithm and mention its robustness in the appendix of the paper.

---

> > ### Comment · Reviewer_5sJt · 2021-09-04
> > **Re: author response**
> >
> > Thank you for your response. I have carefully read through it and the comments from other reviewers.
> >
> > - I appreciate the efforts to add the additional experiments about the nearest neighbour baseline. One suggestion is that it would be better to use L1 norm for the depth version to align with the loss function.
> > - The clarification about the background subtraction step clears up my doubts.
> >
> > I think that the revision so far has improved the quality of the paper.
> > However, as discussed by all reviewers and summarized by the meta-review, the experiments are limited to relatively simple scenarios.
> > The authors argued that the current setup already has its own challenges, which I agree.
> > But after reading the paper and the author response, I do not feel quite confident in that the approach, if extended, would work in more realistic scenarios.
> >
> > As the authors discussed in the challenges faced during the data collection, the method is quite sensitive to the audio noise, and even the humidity.
> > I appreciate that the authors were careful with the data collection. But that makes me wonder whether the approach is practical, especially that it is motivated in a robotics manipulation context. It seems that even for the relatively simple scenarios considered, the method requires a controlled, quiet environment.
> >
> > To summarize, I want to thank the authors for the additional work. The author response did address my concerns 2 and 3. But concern 1 mostly remains.
> > In its current form, this paper is not quite in the “strong accept” category, which is the only higher rating available. Hence my rating remains.

---

### Official Review · Reviewer_XY9t · 2021-07-23

**Originality:** Good
**Technical Quality:** Fair
**Clarity Of Presentation:** Very Good
**Impact:** 4

**Recommendation:**

Weak Accept: I recommend accepting the paper, but will not argue for my recommendation if the majority of other reviewers have a different opinion.

**Summary:**

The paper presents a fun idea: predicting the content of a container attached from 4 low-cost microphones. The vibrations from dropping objects into a container are fed to a convolutional network that predicts the visual reconstruction of the scene (RGB-D output). There's nothings exciting going on in the method, but the application domain is very interesting and seems very much unexplored in the deep-learning robotics era.

**Issues:**

The method is great, but it is severally lacking additional experiments; see 'weaknesses' section for suggested next steps.

**Reviewer Expertise:**

Good: General knowledge of the area

**Strengths And Weaknesses:**

**Strengths:**
- The paper is well written and the problem statement is made clear.
- The method is simple and elegant, and deserves to be built upon.
- The ablation and feature visualisation is interesting and informative.

**Weaknesses**
- The key for an application heavy paper like this, is a solid set of results that proves the claim that this is useful in the wider robotics community for a variety of applications. However, the paper only presents results on 3 wooden blocks. This makes it very difficult to assess how well this method would generalise to a larger number of objects, or if the method would have the capability to generalise to unseen test data given a large diverse training set of objects.
- Section 5.3 is perhaps a little misleading: initially it seems that the paper is presenting results to suggest that they are evaluating on an unknown shape, which (perhaps subjectively) would indicate that the object was not in the training set, however, what they simply mean is that the model is train simultaneously on the 3 shapes, rather than individually training a network for each shape (as in Section 5.4: Reconstruction with known shape).
- The lack of thorough experimentation suggests to me that either the paper did not have enough time to run additional experiments (due to conference deadline) or that additional, challenging setups were run but the method was not successful. Hopefully the former, and I hope the authors use the time after CoRL to continue additional experiments. There are several un answered questions: What about multiple objects in the bin? What if there is already and object in the bin? What is the generalisation capability across unknown objects? What about across containers?

**Summary Of Recommendation:**

In summary, I very much like the paper, and the idea is promising, however the combination of poor technical novelty and poor experimental analysis makes this paper difficult to accept in its current form. I urge the authors to continue working on this line of work and resubmit to a new venue after evaluating the method on a larger dataset (of objects and containers) and assessing generalisation ability atleast across many objects, but also, if possible, across containers.

---

> ### Author Response · Authors · 2021-08-28
> **Responses to Reviewer XY9t (Part 2)**
>
> **“Section 5.3 is perhaps a little misleading: initially it seems that the paper is presenting results to suggest that they are evaluating on an unknown shape, which (perhaps subjectively) would indicate that the object was not in the training set, however, what they simply mean is that the model is train simultaneously on the 3 shapes, rather than individually training a network for each shape (as in Section 5.4: Reconstruction with known shape).”**
>
> Thank you for your suggestion. We have updated the paper to use “mixed shapes” instead of “unknown shapes” to avoid confusion.
>
> **“The lack of thorough experimentation suggests to me that either the paper did not have enough time to run additional experiments (due to conference deadline) or that additional, challenging setups were run but the method was not successful. Hopefully the former, and I hope the authors use the time after CoRL to continue additional experiments. There are several unanswered questions: What about multiple objects in the bin? What if there is already and object in the bin? What is the generalisation capability across unknown objects? What about across containers?”**
>
> As explained above, the current dataset collection serves as a strong real-world challenge. The data collection process involves a large amount of human effort with careful considerations.  Research on multiple objects in the bin, objects already in the bin, and generalization across unknown objects require several generations of research which we feel is beyond the scope of this initial study. We thank the reviewer for the encouragement to keep exploring these directions. We would love to explore the generalization capability under various shapes and more complex scenarios in future works. We will update the paper with these discussions.
>
> **“The combination of poor technical novelty and poor experimental analysis makes this paper difficult to accept in its current form.”**
>
> Our paper is unconventional. While there are numerous studies in the robotics community on using cameras to perceive the state of the world, there are relatively few dedicated investigations into exploiting ambient sound for sensing. Our paper is novel because we developed an integrated solution that combines machine learning with hardware to use natural sound for state estimation.
>
> We argue that presenting a novel setup to leverage a new modality as input is a significant contribution by itself. We have presented the technical challenges of solving this problem with previous methods such as TDoA with engineered features and nearest neighbor methods. Our results show that our solutions achieve much stronger performance. Our solution comes with many novel design choices on both hardware and software beyond a straightforward application. We have also provided a thorough analysis and ablation studies to better understand the learning system, as pointed out by the reviewer. Therefore, we believe that our paper can serve as an important contribution to the CoRL community.

---

> > ### Comment · Reviewer_XY9t · 2021-08-31
> > **Reviewer Response**
> >
> > Thank you for your response. I appreciate that data collection, especially for robotics, can be difficult and time consuming. As I said in my initial review, I do like this paper, and indeed the idea is novel, but I still think the experimental analysis is overall weak; missing any notion of generalisability across objects/containers/etc. My issue is that as a reader, the paper does not give me confidence right now that this will extend beyond the simple setup they have shown, making the usefulness of approach limited. If the paper had a broader set of experiments, it would be an easy strong accept for me. However, due to the uniqueness of the problem domain, I am willing to increase my score to a weak accept.

---

> > > ### Author Response · Authors · 2021-08-31
> > > **Response to Reviewer XY9t**
> > >
> > > We greatly appreciate your valuable recognition and understanding. We definitely look forward to diverse variations on complex scenarios in future works. Thank you!

---

> ### Author Response · Authors · 2021-08-28
> **Responses to Reviewer XY9t (Part 1)**
>
> Thank you for your constructive feedback. We hope that we can clarify and address your concerns with the point response below.
>
> **“The key for an application heavy paper like this is a solid set of results that proves the claim that this is useful in the wider robotics community for a variety of applications. However, the paper only presents results on 3 wooden blocks. This makes it very difficult to assess how well this method would generalize to a larger number of objects, or if the method would have the capability to generalize to unseen test data given a large diverse training set of objects”**
>
> We argue that presenting a novel setup to leverage a new modality as input is a significant contribution by itself. The key contribution of our paper is to present a novel approach to leverage acoustic vibrations for robot perceptions through an integrated hardware and software solution with many novel design choices. We believe that our contribution is beyond a straightforward application.
>
> Scaling the method to a large number of objects under different scenarios is definitely the future step, but we are afraid that it is not doable all at once in our pilot study. We would like to emphasize that the current dataset is already quite challenging to study as is. As shown in our results, traditional sound engineering techniques suffer a lot under the challenging setup (fast motion, chaotic trajectory, small bin with echo, etc. As laid out in Section 3.2.). Even with our learning-based approaches, the scores vary among different shapes with the overall localization percentage being 87%. It requires more research efforts to even solve the current dataset. Therefore, we believe large-scale study on more objects deserves several iterations of future works along the direction. We hope this paper will spur more research on sound inside the robotics community.
>
> Moreover, collecting the dataset with the current setup is not trivial and already reveals useful conclusions for future studies. Specifically, there are several challenges and considerations during the dataset collection:
>
> - The dataset was collected under the restrictions of the COVID-19 pandemic where we do not have a lab environment. To obtain a controlled dataset (more on this below) for rigorous research, we had to overcome multiple challenges to construct the dataset.
>
> - We have collected more than 500 data points for each object shape. In total, there are more than 1,500 data points. This is already non-trivial for real-world data collection which costs about 15 days of data collection, setup inspection, and manual data quality inspection. We have manually inspected all the 4-channel audios, RGB images, as well as depth images.
>
> - Challenges of collecting real-world well-controlled datasets include:
>
>   - The quality of the audio could be affected by the ambient sound. To obtain controlled audio signals, the environment needs to be quiet enough. We also need to consider the impact of the noise made by the dropping motion on other people in the neighborhood, so we could only collect the data within certain hours during quiet periods.
>   - The humidity of the wooden objects might affect the sound. We avoided this by wearing disposable gloves every time we started the data collection.
>   - To avoid the bias of lightning conditions on the ground truth RGB-D images, we needed to control the lightning conditions.
>   - The table surface holding the bin needs to be maintained at the same level to avoid potential bias introduced by the table level. We carefully calibrate the table surface every time before data collection. The cameras also need to be fixed to avoid bias introduced into the dataset.
>   - The color of the wooden objects may fade over a long time period, so the data collection has to happen within a short timeframe.
>   - Other challenges could affect the stationarity of the data collection: the bin might deform over time due to dropping motions, mics contact properties might change over time, the bin might move during impact. All of these need to be verified during the collection process.
>   - There is a crucial factor: the fact that humans need to drop the object to simulate real-world scenarios limits the possibility to automate the data collection.
>
> We have provided detailed analysis and ablation studies to carefully understand the overall process. As shown in the results, even the current dataset can break the traditional baseline methods and it is also not completely solved by the learning-based methods. Moreover, the direction we are studying is relatively underexplored, we believe that our research can serve as an important reference for future works.

---

### Official Review · Reviewer_Ezuu · 2021-07-26

**Originality:** Good
**Technical Quality:** Good
**Clarity Of Presentation:** Good
**Impact:** 3

**Recommendation:**

Weak Accept: I recommend accepting the paper, but will not argue for my recommendation if the majority of other reviewers have a different opinion.

**Summary:**

The paper predicts the rgb and depth of an object in a bin given the audio signals from contact microphones attached to the bin. The audio signals capture the sounds/vibrations caused by the object when it’s dropped into the bin.

**Issues:**

To reiterate, some things to be done to improve the paper:
1. Needs experiments across more objects.
2. Show application in a downstream task such as moving the object to the center of the bin.


**Reviewer Expertise:**

Good: General knowledge of the area

**Strengths And Weaknesses:**

The paper’s motivation is very promising and could lead to better use of audio for robotic manipulation. However, there are several concerns with the paper which are listed below:

1. The Localization error metric only counts the number of times the two bounding boxes overlap by a certain distance. Adding an additional metric that computes the distance between the two bounding boxes will be useful. Specifically capturing  both position accuracy and orientation accuracy. Without a measure of accuracy in meters/radians, it’s not clear how accurate the method will be for use in robot manipulation.

2. How will the predictions scale to more objects? E.g., Can the method differentiate between a ball and a cube where the ball could move around a lot before settling? Can the method differentiate between sharp edged objects (star?) and soft edged objects?

3. It’s unclear if the accuracy of the predictions are sufficient for manipulation tasks. Using the prediction in a downstream task and evaluating task success across the different methods is needed.

4. Can these predictions be generated online? This is especially important for feedback control.


**Summary Of Recommendation:**

Mentioned weaknesses need to be addressed to show the benefit of the proposed approach.

---

> ### Author Response · Authors · 2021-08-28
> **Responses to Reviewer Ezuu (Part 3)**
>
> **“It’s unclear if the accuracy of the predictions is sufficient for manipulation tasks. Using the prediction in a downstream task and evaluating task success across the different methods is needed.”**
>
> As mentioned above, our focus is to demonstrate a novel hardware and software setup for the perception module. Evaluating the performance of downstream tasks involves additional efforts on robot planning and control. Different tasks require various evaluation metrics. Therefore, connecting our proposed method with downstream robotic tasks is out of scope.
>
> **“Can these predictions be generated online? This is especially important for feedback control.”**
>
> Yes, the predictions can be generated online in a real-time fashion. Since our approach only requires one forward pass of a lightweight neural network on a single GPU, the predictions can be obtained with 188 Hz on a laptop.

---

> ### Author Response · Authors · 2021-08-28
> **Responses to Reviewer Ezuu (Part 2)**
>
> **“How will the predictions scale to more objects? E.g., Can the method differentiate between a ball and a cube where the ball could move around a lot before settling? Can the method differentiate between sharp-edged objects (star?) and soft-edged objects?”**
>
> Thank you for your great suggestions. However, we would like to emphasize that the current dataset is already quite challenging to study as is. As shown in our results, traditional sound engineering techniques suffer a lot under the challenging setup (fast motion, chaotic trajectory, small bin with echo, etc. As laid out in Section 3.2.). Even with our learning-based approaches, the scores vary among different shapes with the overall localization percentage being 87%. It requires more research efforts to even solve the current dataset. Therefore, we believe large-scale studies on more objects deserve several iterations of future works along the direction. We hope this paper will spur more research on sound inside the robotics community.
>
> Since our approach takes vibrations of the bin as input, we believe our method can still work well with sharp-edged objects and soft-edged objects because they will have different sound profiles from their trajectories. Similarly, cubes and spheres make quite different sounds throughout the motion. Learning to leverage the acoustic signatures of various objects is the underlying mechanism of our approach and it is also the strength of our approach assuming appropriate training on similar distributions.  From our data collection, we have found that the contact microphones are quite sensitive to small perturbations to pick up subtle but critical signals. As the sensors do not require calibration and are of low cost, more contact microphones can be placed around the bin to obtain richer signals.
>
>  A crucial advantage of our paper is that we work with real data in the physical world. But, this also means collecting this real-world dataset is extremely challenging:
>
> - The dataset was collected under the restrictions of the COVID-19 pandemic where we do not have a lab environment. To obtain a controlled dataset (more on this below) for rigorous research, we had to overcome multiple challenges to construct the dataset.
> - We have collected more than 500 data points for each object shape. In total, there are more than 1,500 data points. This is already non-trivial for real-world data collection which costs about 15 days of data collection, setup inspection, and manual data quality inspection. We have manually inspected all the 4-channel audios, RGB images, as well as depth images.
> - Challenges of collecting real-world well-controlled datasets include:
>
>   - The quality of the audio could be affected by the ambient sound. To obtain controlled audio signals, the environment needs to be quiet enough. We also need to consider the impact of the noise made by the dropping motion on other people in the neighborhood, so we could only collect the data within certain hours during quiet periods.
>
>   - The humidity of the wooden objects might affect the sound. We avoided this by wearing disposable gloves every time we started the data collection.
>   - To avoid the bias of lightning conditions on the ground truth RGB-D images, we needed to control the lighting conditions.
>   - The table surface holding the bin needs to be maintained at the same level to avoid potential bias introduced by the table level. We carefully calibrate the table surface every time before data collection. The cameras also need to be fixed to avoid bias introduced into the dataset.
>   - The color of the wooden objects may fade over a long time period, so the data collection has to happen within a short timeframe.
>   - Other challenges could affect the stationarity of the data collection: the bin might deform over time due to dropping motions, mics contact properties might change over time, the bin might move during impact. All of these need to be verified during the collection process.
>   - There is a crucial factor: the fact that humans need to drop the object to simulate real-world scenarios limits the possibility to automate the data collection.

---

> > ### Comment · Reviewer_Ezuu · 2021-09-04
> > **Thanks [increased score]**
> >
> > Thank you for the answers. I understand the hardship in collecting this dataset and maybe a follow-up work on how to collect these datasets with little effort would be interesting to read. I think generalization is a major unknown with the current results. However, I have upgraded my score since this is a novel domain for robotics.

---

> ### Author Response · Authors · 2021-08-28
> **Responses to Reviewer Ezuu (Part 1)**
>
>
> Thank you for your constructive comments and suggestions. We will try our best to address your points below:
>
> **“The Localization error metric only counts the number of times the two bounding boxes overlap by a certain distance. Adding an additional metric that computes the distance between the two bounding boxes will be useful. Specifically capturing both position accuracy and orientation accuracy.”**
>
> We would like to clarify that our evaluations consist of two metrics, which do evaluate your suggested metric. The first one is the localization metric which evaluates the precision of the predicted object center. For suction-based manipulators, this evaluation metric provides a good amount of estimation of the potential downstream task performance. For other manipulators where both the position and orientation matter as suggested by the reviewer, we presented a second intersection over union (IoU) metric. A high IoU score requires a good predicted shape, position, and orientation. These two metrics construct a sufficient evaluation pipeline for our perception algorithms.
>
> To rigorously evaluate downstream tasks' performance, various tasks would require different evaluation metrics. For example, “moving the object to the center of the bin”, requires more understanding of the object dynamics under a quasi-static situation that needs to be evaluated with goal precision. Another example is to evaluate the success rate of object retrieval by picking the object up. All involve the control and planning process that goes beyond the perception module. However, this goes beyond the current scope of the paper where we only study a novel perception method with audio modality. We, therefore, leave the downstream applications as future work.

---

### Meta-Review · Area_Chair_a6US · 2021-08-11

**Recommendation:** Accept (Poster)
**Confidence:** 4

**Metareview:**

This paper presents a new hardware setup for estimating the shapes and poses of objects inside a bin using only auditory signals. The objects inside the bin are bounced and the sounds resulting from their collisions are captured by four microphones. The signals from the microphones are used as inputs to a convolutional neural network that returns an RGB image of the objects inside the bin. The proposed method is evaluated on scenes containing single wooden cubic objects. The experimental results clearly indicate that the proposed method is able to reconstruct the interior of the bin.
The reviewers agree that the proposed method is interesting and novel. The reviewers find that the main contribution of this work is the application, rather than the technical approach. Therefore, the paper should include a richer and more exhaustive set of experiments. The current ones are limited to single objects that have simple regular shapes. The reviewers are not sure why the paper does not include experiments on realistic objects, or multiple objects in the bin. The evaluation metrics can also be improved. Finally, stronger baselines should be considered. For example, the authors can try simple nearest-neighbors methods, or traditional shallow learning techniques.

In their rebuttal, the authors added new experiments and addressed several concerns of the reviewers and the AC. From the discussion, it is clear that the reviewers appreciate the novelty of this application. However, concerns regarding the generalization and practicality of the proposed technique remain. Therefore, the reviewers are leaning toward a weak accept. In conclusion, this is a good paper that deals with a challenging problem where it is difficult to collect more data. The area chair believes that this is a promising research direction, and this paper may open the door to more elaborate works in this new domain.

---

> ### Author Response · Authors · 2021-08-28
> **Responses to Area Chair a6US**
>
> Thank you for your appreciation of our work and constructive feedback.
>
> **Novelty and Contribution**
>
> We would like to emphasize that our paper is unconventional. While there are numerous studies in the robotics community on using cameras to perceive the state of the world, there are relatively few dedicated investigations into exploiting ambient sound for sensing. Our paper is novel because we developed an integrated solution that combines machine learning with hardware to use natural sound for state estimation.
>
> We argue that presenting a novel setup (hardware and learning algorithms) to leverage a new modality as input is a significant contribution by itself. We have presented the technical challenges of solving this problem with previous methods such as TDoA with engineered features and nearest neighbor methods. Our results show that our solutions achieve much stronger performance. Our solution comes with many novel design choices on both hardware and software beyond a straightforward application. We have also provided a thorough analysis and ablation studies to better understand the learning system, as pointed out by the reviewers. Therefore, we believe that our paper can serve as an important contribution to the CoRL community.
>
> **Responses to Reviewers and New Experiments**
>
> We appreciate the suggested nearest neighbor baseline, and our new experiments (Figure 6) suggest that our conclusion still holds. We updated the paper to include these new experiments and attached our responses to all the reviewers.
>
> We gave extensive responses with detailed explanations and additional experiments to each individual reviewer. We hope that our responses help resolve the reviewer’s concerns.
>
> **Dataset**
>
> Regarding the scale of the dataset. We would like to emphasize that the current dataset is already quite challenging to study as is. As shown in our results, traditional sound engineering techniques suffer a lot under the challenging setup (fast motion, chaotic trajectory, small bin with echo, etc. As laid out in Section 3.2.). Even with our learning-based approaches, the scores vary among different shapes with the overall localization percentage being 87%. It requires more research efforts to even solve the current dataset. Therefore, we believe large-scale studies on more objects deserve several iterations of future works along the direction. We hope this paper will spur more research on sound inside the robotics community.
>
> Moreover, collecting the dataset with the current setup is not trivial and already reveals useful conclusions for future studies. Specifically, there are several challenges and considerations during the dataset collection:
>
> - The dataset was collected under the restrictions of the COVID-19 pandemic where we do not have a lab environment. To obtain a controlled dataset (more on this below) for rigorous research, we had to overcome multiple challenges to construct the dataset.
>
> - We have collected more than 500 data points for each object shape. In total, there are more than 1,500 data points. This is already non-trivial for real-world data collection which costs about 15 days of data collection, the setup inspection, and manual data quality inspection. We have manually inspected all the 4-channel audios, RGB images, as well as depth images.
>
> - Challenges of collecting real-world well-controlled datasets include:
>   - The quality of the audio could be affected by the ambient sound. To obtain controlled audio signals, the environment needs to be quiet enough. We also need to consider the impact of the noise made by the dropping motion on other people in the neighborhood, so we could only collect the data within certain hours during quiet periods.
>   - The humidity of the wooden objects might affect the sound. We avoided this by wearing disposable gloves every time we started the data collection.
>   - To avoid the bias of lightning conditions on the ground truth RGB-D images, we needed to control the lighting conditions.
>   - The table surface holding the bin needs to be maintained at the same level to avoid potential bias introduced by the table level. We carefully calibrate the table surface every time before data collection. The cameras also need to be fixed to avoid bias introduced into the dataset.
>   - The color of the wooden objects may fade over a long time period, so the data collection has to happen within a short timeframe.
>   - Other challenges could affect the stationarity of the data collection: the bin might deform over time due to dropping motions, mics contact properties might change over time, the bin might move during impact. All of these need to be verified during the collection process.
>   - There is a crucial factor: the fact that humans need to drop the object to simulate real-world scenarios limits the possibility to automate the data collection.

---

### Decision · Program_Chairs · 2021-09-13

**Decision:**

Accept (Poster)

**Comment:**

This paper presents a new hardware setup for estimating the shapes and poses of objects inside a bin using only auditory signals. The objects inside the bin are bounced and the sounds resulting from their collisions are captured by four microphones. The signals from the microphones are used as inputs to a convolutional neural network that returns an RGB image of the objects inside the bin. The proposed method is evaluated on scenes containing single wooden cubic objects. The experimental results clearly indicate that the proposed method is able to reconstruct the interior of the bin.
The reviewers agree that the proposed method is interesting and novel. The reviewers find that the main contribution of this work is the application, rather than the technical approach. Therefore, the paper should include a richer and more exhaustive set of experiments. The current ones are limited to single objects that have simple regular shapes. The reviewers are not sure why the paper does not include experiments on realistic objects, or multiple objects in the bin. The evaluation metrics can also be improved. Finally, stronger baselines should be considered. For example, the authors can try simple nearest-neighbors methods, or traditional shallow learning techniques.

In their rebuttal, the authors added new experiments and addressed several concerns of the reviewers and the AC. From the discussion, it is clear that the reviewers appreciate the novelty of this application. However, concerns regarding the generalization and practicality of the proposed technique remain. Therefore, the reviewers are leaning toward a weak accept. In conclusion, this is a good paper that deals with a challenging problem where it is difficult to collect more data. The area chair believes that this is a promising research direction, and this paper may open the door to more elaborate works in this new domain.